# A Survey on Gamification for Health Rehabilitation Care: Applications, Opportunities, and Open Challenges

**Nooralisa Mohd Tuah** [1,*] **, Fatimah Ahmedy** [2] **, Abdullah Gani** [1] **and Lionelson Norbert Yong** [2]

[1] Faculty of Computing and Informatics, Universiti Malaysia Sabah, Kota Kinabalu, Sabah 88400, Malaysia; abdullahgani@ums.edu.my
[2] Faculty of Medicine & Health Sciences, Universiti Malaysia Sabah, Kota Kinabalu, Sabah 88400, Malaysia; fatimahmedy@ums.edu.my (F.A.); lionelson95@gmail.com (L.N.Y.)
[*] Correspondence: alisa.tuah@ums.edu.my

**Abstract:** Research trends in gamification have shown a significant diversity in various areas of e-health, particularly in addressing the issues of rehabilitation and physical activity. Rehabilitation requires better engaging tools that help to increase the patient's motivation and engagement in particular forms of rehabilitation training. Adopting gamification in rehabilitation offers different treatment and care environments when implementing rehabilitation training. As gamification is increasingly being explored in rehabilitation, one might not realize that using various techniques in gamified applications yields a different effect on gameplay. To date, varied gamification techniques have been utilized to provide useful experiences from the perspective of health applications. However, a limited number of surveys have investigated the gamification of rehabilitation and the use of suitable game techniques for rehabilitation in the literature. The objective of this paper is to examine and analyze the existing gamification techniques for rehabilitation applications. A classification of rehabilitation gamification is developed based on the rehabilitation gamifying requirements and the gamification characteristics that are commonly applied in rehabilitation applications. This classification is the main contribution of this paper. It provides insight for researchers and practitioners into suitable techniques to design and apply gamification with increased motivation and sustainable engagement for rehabilitation treatment and care. In addition, different game elements, selection blocks, and gamification techniques are identified for application in rehabilitation. In conclusion, several challenges and research opportunities are discussed to improve gamification deployment in rehabilitation in the future.

**Keywords:** gamification; rehabilitation; e-health

## 1. Introduction

Promoting a healthy lifestyle is a huge challenge for some practitioners and healthcare institutions, especially when the delivery method is less interactive and focuses on one-way communication. Shifting the delivery methods from traditional to digitalizing, playful, and interactive modes could bring a more favorable outcome in the rehabilitation process, as well as in individual changes of behavior. In the past decade, many interactive methods have been introduced for rehabilitation training. These methods include the use of serious games and gamification during rehabilitation therapy [1,2], physical activity monitoring [3], health promotion [4], and rehabilitation virtual learning apps [5]. The use of gamification is seen as a motivational driver to enhance individual engagement towards the rehabilitation process and, thus, reduce hospitalizations and encourage self-management. In recent years, research related to gamification in rehabilitation training has generated considerable research interest. It plays a vital role in promoting the effectiveness of the recovery process during rehabilitation and inducing efficacy from the perspective of healthcare management.

In general, researchers have recognized gamification as using the game elements or game mechanics in activities that do not represent a gaming context, such as learning,

teaching, and healthcare, while preserving the playfulness of the environment [6,7]. Meanwhile, rehabilitation is a process for an individual with a health condition to return to a healthy life or one that helps them to improve abilities that they need in daily life through a series of restoration processes such as training and therapy [1]. This process requires continuous engagement and involvement from the affected individual, and it sometimes causes stress, dropout, and unmanageable care. Therefore, introducing gamification into rehabilitation training helps to prevent those situations from occurring.

Much research in recent years had focused on gamification in e-health and wellbeing (see, for example, studies by [8,9]). These works provide reviews of the existing solutions based on gamified applications in medical informatics, as well as analyses of the impact of the solution on individual health. Research by Sardi et al. [9] reported that the rehabilitation field was among the most-investigated health domains regarding gamification and serious games. The review shows that the use of gamification is generally accepted with increasing implementation in real healthcare practice. Among the investigated rehabilitation domains that used gamification as part of rehabilitation, interventions include post-stroke, chronic pain, and physical injuries [1–3,10]. For example, research on post-stroke conditions by Tamayo-Serrano et al. [10] and Polak et al. [11] embedded game elements, such as levels and points, in each of the therapy activities in their gamified system, whereby the players collected points for making a correct movement. Similarly, in a study by Afyouni et al. [5] on hand injuries, game elements such as stars, trophies, and badges were utilized as a reward system. The reward was given when the person had successfully conducted the required hand movement exercises within the given environment. This reward system is one of the gamification techniques used to encourage individuals' participation in rehabilitation training. Generally, gamification technique is a building block of game elements designed together to aim at specific experiences or targeted behavior changes in gameplay. Those applications generally used basic game elements that aimed to encourage individuals' participation in rehabilitation training.

Although many gamification techniques were applied in gamified rehabilitation applications, they mainly focused on specific rehabilitation domains and techniques. There are review articles that explored the gamification techniques and strategies used in health [8,11] and in stroke rehabilitation [10]. These reviews addressed different aims, domains, and benefits of gamification research. However, there remains a need to explore state-of-the-art developments on the various gamification applications that sustain individual engagement and involvement in rehabilitation. Additionally, there is a lack of specific guidance to assist researchers in comparing or selecting suitable applications of gamification techniques and the type of interventions used for gamifying the rehabilitation. Thus, a survey consisting of comparative studies in this particular focus is necessary.

This paper aims to survey recent gamification applications in rehabilitation, which leads to developing a classification of the gamified applications in rehabilitation, including both mobile applications and computer systems. This paper provides a study on state-of-the-art gamification applications in rehabilitation with the primary objective of yielding a comprehensive classification. For that, this paper will attempt the following:

(a) To identify and evaluate gamification applications used in rehabilitation;
(b) to classify the gamification applications used in rehabilitation; and
(c) to discuss significant challenges of the gamification applications in rehabilitation and determine the potential research opportunities in e-health.

The structure of this paper is organized as follows: Section 2 presents the methodology employed in conducting the survey. Section 3 begins with the explanation of the gamified application for rehabilitation, which involves the surveys related to the gamification terms, concepts, and game elements applied for rehabilitation. Section 4 continues with an elaboration of current gamification trends in rehabilitation followed by a discussion on the requirements for gamifying rehabilitation. Section 5 presents the types of intervention in rehabilitation, and the mapping between the need and the intervention. Section 6 discusses the selections of a game elements building block. Following the discussion, the section

presents the classification of gamification applications in rehabilitation. In Section 7, the focus is on the challenges of applying gamification in rehabilitation. Moreover, potential research opportunities are addressed. Finally, the concluding remarks and the significance of the study are provided in Section 8.

## 2. Survey Methods

A survey of the literature was conducted to provide a comprehensive analysis and synthesis of the available studies in the particular research area. In this paper, the survey was conducted on the concept of gamification and its application in rehabilitation. A rigorous search of the literature was undertaken in the subject areas using the databases PubMed, ACM, IEEE Xplore, Scopus, and ScienceDirect. This selection of databases was based on the multidisciplinary nature of human-computer interaction research, computer applications, biomedical application, and games. A comprehensive search using the search queries (("gamification OR gamified OR gamify") AND ("application OR system OR web") AND ("rehabilitation OR rehab OR telerehabilitation OR exercise OR therapy")) and reviewing sources such as relevant types of journal articles and conference material from 2010 to 2020 generated a collection of 294 articles. A ten-year period of publication was considered in this survey because, through searching, the concept of gamification was explored and receiving more attention from 2010. The database was searched on 20 June 2020 and the result is shown as follows: ACM ($n = 70$), IEEE Xplore ($n = 97$), PubMed ($n = 32$), Scopus ($n = 50$), and ScienceDirect ($n = 45$).

Among these, only articles specific to the gamification application in rehabilitation were included, and these articles had to be an implementation paper, not a conceptual or theoretical paper. The selection of articles was made by the authors based on several inclusion criteria. They are (a) the application of gamification is clearly design and implemented; (b) empirical research was conducted whereby the experiment and findings are reported; and (c) the design and purpose of rehabilitation are clearly defined. An initial 53 articles were considered. However, three papers were inaccessible due to limited access, five papers were in a workshop article, and the papers were relatively similar. Another three papers were repetitive as they were an on-going work, in that they were using the same gamified application. Thus, only the latest articles were included. Based on all criteria, a final 42 articles were included in this survey. Figure 1 illustrates the process of article's inclusion in this study.

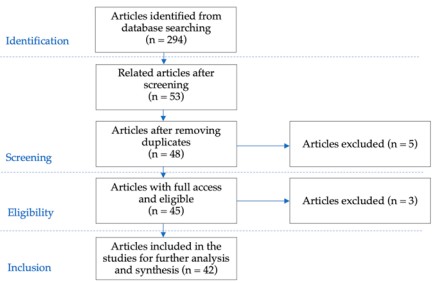

**Figure 1.** Process of article inclusion.

## 3. Gamification and Rehabilitation

Recent research in e-health has suggested that rehabilitation is among the most common health condition that utilizes gamification as one of the intervention tools. The applications have shown different implementations of gamification techniques in improving rehabilitation outcomes. This difference implied great potential and room for improvement in gamification, and for its performance to be enhanced and expanded. The use of gamification techniques serves as a platform for promoting individuals' motivation to engage and participate in rehabilitation training more interactively. In this section, the gamification of rehabilitation is elaborated. The concepts of gamification are elaborated, followed by the game elements that are designated in a gamified application.

### 3.1. The Concepts of Gamification

In the past decade, gamification has evolved significantly within the healthcare setting. It has gained considerable interest in research as well as in practice. The concept of gamification is not new, and since its beginning, the idea has emerged in, and been adapted by, various domains. The use of gamification has proven to have positive effects due to several factors, including game experiences, game environments, personal analytics on progression and data tracking, and affordable technology [6,9]. These factors have indirectly led to more exploration of its application in e-health. Thus, this highlights an increase in the body of scholarship on the related topic. The need to understand gamification and its utilization has motivated a move towards user engagement within human-computer interaction (HCI) [6] and in the direction of user changes in behavior within the health context [8]. Deterding et al. [6] defined it as "the use of game design elements in non-game contexts" to understand gamification. The terms imply that game designs, elements, mechanics, and procedures are used in an application, a system, services, or products that are not in a gaming environment but are rendered into a more game-like environment. Deterding et al. [6] suggested that the application of gamification should involve the elements of gamefulness, gameful interaction, and gameful design. In this context, a player should perceive a live experience (gamefulness) gained while interacting with the objects or be in the context of the application (gameful interaction), and this requires a gameful design to evoke live experiences [6,12].

Gamification has also been conceived as a process of creating a system or application with game-like experiences [13]. In this view, gamification should be thought of as a process that emphasizes more the creation of an application while enhancing the game-like aspects of the design. Besides, the process should not be about how points or badges are applied in the gamified application. Huotari and Hamari [14] advocate that gamification, in a broad understanding, can be described as a process that increases the possibility of a user or a player developing gameful experiences by instilling in the application or system particular game elements. Here is where gamefulness, gameful interaction, and gameful design play roles in crafting a gamification application. In the application, a few selections of game elements are combined and applied interactively in a system. However, with so many points contained in the gamification concept (see studies by [6,7,9,13,15]) and how it has been differentiated from games and serious games, one should consider that the gamification application might not act fully as an operational game, but instead as an end product [9,12].

Meanwhile, gamification in e-health is plausibly understood as a persuasive technology-based serious game with embedded personal informatics that positively motivates specific behavior and experiences [8]. It aims to drive particular behavior through a well-designed game element and the implementation of the techniques. Research has suggested that a specific combination of game elements that are designated together could induce changes in individual behavior [8,9]. Hence, comprehending the available game elements will help to design a suitable application for rehabilitation.

### 3.2. General Game Elements

Game elements are the essential elements, or a set of building blocks or features of games, designated in gamification applications [13,16]. Following Hamari et al. [7], the game elements are generally the 'affordance' for a gameful experience in which the game elements are considered the designed stimulus that incites a user's motivational needs. Thus, there is a possibility of recurring experiences. Several basic game design elements have been implemented in gamification applications in established studies, and these elements are summarized based on the review research by [7–9,12]. Despite the overlaps among the studies detected from the list, different design selections of the building blocks have been implemented and, in some ways, have become the characteristics of the gamified application. However, this is not the only list of game elements. Here, a comprehensive list of game elements that have been used in gamified applications is delivered. These game

elements are occasionally used with different names, but the features remain the same. The game elements are as follows:

1.  Points—The game element is an essential one that is typically used as a reward for completing particular activities in the gamified environment. A gamified application can adopt several types of points depending on the purpose the points serve. Points are collected, transferred, and reused for other purposes. This game element is implemented as a reward for gaining experiences, obtaining a higher or different status; or for providing feedback; or as redeemable points. Alternatively, points are also recognized as rewards and scores.

2.  Leaderboard—The leaderboard is an element that summarizes and shows a player's achievement. The leaderboard is an indicator of player performance where it might induce among the players a desire to compete. It shows the players' achievements and ranks them based on their best performance in a particular task or activity, or in the gameplay overall.

3.  Badges—Badges are a symbol of status. A badge is an element that symbolizes a specific achievement in gamified applications. Badges also are collected after earning a particular number of points. Generally, badges are not compulsory; however, receiving them can socially influence players' behavior, mainly when the badges are rare and difficult to get. Every achievement (completing levels or goals) will mean receiving different badges that show different statuses of players in the gameplay. Alternatively, it is also recognized as a medal.

4.  Trophies—A trophy is a specific award given to a player upon the accomplishment of a particular task or activity. Unlike points and badges, trophies usually relate to the achievement of behavior. They are awarded when a player achieves a behavioral goal in gameplay. For example, every person's health records in a gamified application might be given points. When a person is logging health records every day for one week, he/she will receive a 'compliance' badge. Achieving a good health record in a week will allow one to get a 'healthy player' trophy. Alternatively, it is also recognized as an award, medal, or stars.

5.  Level—The level is an element applied based on points, where the players can move from one level to another level as they reach a certain required stage. Levels are used to indicate game difficulties. A higher level is usually installed with increasingly difficult challenges.

6.  Ranks—A ranking positions the players from the top to the lowest. It is presented to all players, and thus, it would increase competitiveness. The roles are just like the leaderboard; however, the latter has more information than the ranking. Player positions are customarily based on their accumulative points and achieved levels. They also are positioned based on their progress in completing the given activities.

7.  Story/theme—A story or theme in a gamified application brings a gameful experience. In some situations, it creates an immersive experience that helps the players to adapt to their situation and learn from it. However, stories play an important role in gamified applications as they create an interesting situation when the stories reflect the player's personal interest. The context or storyline is an analogy of a real-world setting. It is also known as the narrative, context, or storytelling.

8.  Feedback—This element provides immediate feedback on the players' performance and achievement of given tasks. Iterative visual feedback is given on the interfaces or through agents. Feedback is also regarded as a response.

9.  Progression—Progression represents milestones. It makes the players alert to their situation and current position. Additionally, it helps the player to strategize their movement so that it will affect their progression. It provides the player's game status or achievement status or how far they achieved them. In some situations, the status bar is used to show the progress.

10. Challenge—An element with increasing difficulties at different levels, this has to be carefully designed as over-easy tasks make the players bored, while over-difficult tasks make the players leave. Alternatively, this element is also known as a quest.

11. Roles/Avatar—An avatar can be identified as the player's virtual identity or self-representation in gameplay. It creates the player roles that evoke immersive environments. Avatars also represent the other characters (non-player characters) who are interactively involved in the game. In many e-health applications, the avatar is used for visual feedback or in response to an individual health condition. It is also known as self-representation and virtual identity

12. Status—A status is an element that shows the player's status in gameplay. It also relates to the ranking that shows a player's reputation. The application is presented through either the player's avatar or badges. It is also known as reputation.

13. Voting—Voting is an element provided to facilitate voting activities. The players will be able to vote and receive responses or results after voting. The players are also offered some in-game rewards like points, bonuses, or stars to motivate them to vote. Voting is generally used to get feedback from the players, or it is just an added fun element in the gameplay.

14. Betting—Betting is an element that enables the players to make a bet on a specific occasion, such as an estimation. The betting winner will receive an in-game reward, like points that are exchangeable for other elements in the gameplay.

By grasping the concept of gamification and its elements, surveying the state-of-the-art developments helps to understand further its application in, as well as the requirements of, rehabilitation.

## 4. Implementing Gamification in Rehabilitation

Implementing gamification has thorough design requirements based on the needs of gamification for rehabilitation. Understanding the gamification needs is a great challenge in enhancing patient's engagement in therapy and care. In particular, the challenges lie in applying suitable techniques in the application. In this section, surveys of current gamification trends in rehabilitation are presented. Considering the trends, this section further elaborates the necessary requirements for the application of gamification in rehabilitation.

### 4.1. Current Gamification Trends in Rehabilitation

The rehabilitation process has proven to be a practical approach to restoring individual self-sufficiency and the abilities needed for performing mobility functions and daily living activities. The use of gamification as a technology-based intervention in e-health is conceptualized as persuasive technology, serious games, and personal informatics [8,17]. This concept revolves around specific purposes and outcomes targeted in e-health. Like in e-health, for rehabilitation training, persuasive technology can be seen as an application that uses the particular design of game elements to encourage specific individual behavior and experiences [3]. For example, previous studies by [3–5] demonstrated that the gamification applications had simulated exercise experiences through the use of levels, whereby the levels were implemented with increasing difficulty. Generally, in the study, the exercises were simulated to encourage individuals to manipulate their hands for the various simulated environments, which served as the path to promote neurological recovery from a hand injury. As for serious games, gamification usage drives individual intrinsic motivation to sustain engagement with the training voluntarily. In many situations, the gamification concept embedded mostly in serious games focused on familiarizing an individual with their conditions and self-care (see example in [4,5]).

Meanwhile, for personal informatics, gamification is generally connected to the elements of progression, leaderboard and feedback, that demonstrate analytics of individual behavior during rehabilitation training [8,18]. In some health applications, such as Strava and Fitbit, the information is presented in the dashboard whereby individuals have the ability to explore their essential information in a single interface. Indeed, based on research

in the field of rehabilitation, informatics is seen as one of the most commonly applied methods used to sustain individual engagement in training [5,18].

Promising gamification applications as part of rehabilitation training intervention requires consideration of the rehabilitation process, procedures, the requirements of gamified interventions in rehabilitation, and its effectiveness, particularly in motivating and encouraging individuals towards high engagement in the intervention with an impetus for an individualized rehabilitation regime. In general, rehabilitation training can be categorized into cognitive and physical [1,10]. The delivery of rehabilitation training has been conducted as home rehabilitation, telerehabilitation, or hospital-based rehabilitation. In addition, gamification has increasingly been applied in mobile-based applications [19–21], virtual reality (VR) or augmented reality (AR) applications [22–24], web-based applications [25], or social applications [26].

Most gamification applications used in rehabilitation practice that existed and were studied in the literature are primarily in the category of physical rehabilitation. Grounded in the selected studies for this survey, the primary information extracted is based on (1) the type of application, (2) the evaluations study, (3) reviewed works, and (4) gamification model or framework development. The research articles are mainly distributed into two domains, physical rehabilitative therapies (88%) and psychological rehabilitation (12%). Of those related to the research in physical treatments, rehabilitation domains investigated include motor-rehabilitation (16%), postural stability (2%), physiotherapy (5%), exercising (7%), hand therapy (17%), heart disease (5%), stroke-related conditions (16%), and general neurological diseases (20%) (such as dysphonic, hemiparesis, Parkinson, cerebral palsy, and concussion). The article distribution is shown in Figure 2. These articles were empirically evaluated through surveys, experiments, and case studies. Further insights on the gamification of rehabilitation training in these researched articles are listed in Appendix A, as a general overview of the domains, the application of gamification for rehabilitation, and the overall outcomes of this research.

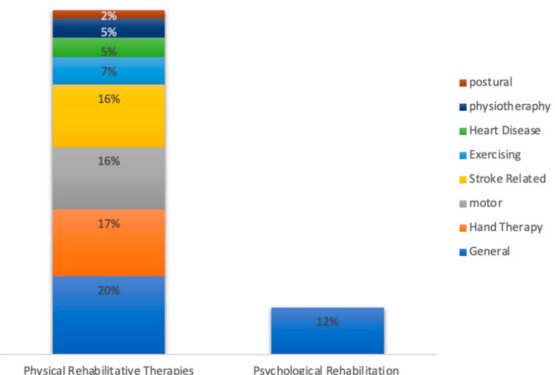

**Figure 2.** Article distribution based on the most gamification applications used in rehabilitation practice.

Based on the information retrieved in Appendix A, three main themes were observed of gamification in rehabilitation training: (1) external devices involved in the application; (2) domain where the gamification is customarily applied; and (3) the findings and outcome from the application. For the first theme, gamification in rehabilitation is mainly involved with external devices such as human-robotic, motion sensors, and mixed reality devices. These external devices are used to support gamified activities, such as exercising, training, physio, and a trainer/companion in rehabilitation care, as a means to facilitate the individuals' performance of the required tasks through the rehabilitation processes. For the second theme, the applications are mainly adopted in several rehabilitation domains. These were categorized in three main areas:

(1) Physiotherapy, which includes motor-rehabilitative, postural stability, exercising, hand therapy, and heart disease;

(2) neurological, which includes stroke-related and neurological disorders; and

(3)    psychological rehabilitation.

The third theme looks at the outcomes of using the applications. Outcomes were identified through the purposes of the gamification implementation and the expected results of the evaluation study. For that, the outcomes were grouped into three categories: motivation for, and engagement in, the care; promising individual behavior; and improvement in health outcomes.

The applications and research summarized in Appendix A are mainly directed to physiotherapy, mostly for guiding the physical movement. The primary rationale for gamifying a physiotherapy process is the nature of the treatment and care. The treatment and care for physiotherapy may be simulated in a straightforward way using a gamified application. In addition, the ability of gamified applications to motivate individuals in rehabilitation training by providing a pleasant and engaging environment during the process has made them suitable for adaptation. In the next section, the rehabilitation gamification application requirements are discussed to provide further insight into how an application can be designed to maximize its function.

*4.2. Rehabilitation Gamifying Requirement*

Motivation and engagement are the basic elements of measuring the effectiveness of a gamification application in rehabilitation training. Identifying the characteristics of a gamification technique is essential to establish rehabilitation requirements, as well as to design and develop the gamified application. To successfully implement gamification, three main design challenges are considered as the requirements, namely gamefulness, gameful interaction, and gameful design [6,12]. These are defined as follows:

- Gamefulness—or a gameful experience, a unique condition [14] arising from the application that brings lived experiences [12] and the behavioral quality of playing [6]. These experiences are the result of a combination of game elements [27].
- Gameful interaction—is an approach of crafting the game elements that bring the gameful experiences in gameplay [6]. This approach also includes the contexts, game objects, or game tools in the gamification application [12].
- Gameful design—The practice or processes to design a gameful gamification application that evokes live experiences [12]. This process involves structuring the tasks and activities with suitable game elements and their techniques.

These requirements should be incorporated in a balanced way when designing and developing the gamification application. This balance ensures that gamification does not solely focus on the game elements, but also the ability to evoke game-like experiences in the application. As these requirements are the most relevant aspects of designing a gamified application, Seaborn and Fels [12] have argued that they had to be further elaborated in the game elements to achieve an enhanced gameful experience. Huotari and Hamari [14] advocated gameful experience as an essential psychological factor linked to gamification's successful implementation. Eppmann et al. [27] adopted these gamefulness design challenges in proposing a measurement scale for gameful experiences in gamification. Therefore, it is vital to gain a certain degree of insight into how these designing challenges should be incorporated into gamified applications.

The design challenges should be considered when crafting and applying gamification techniques in a gamified application. These challenges will be mapped onto the game elements and the techniques used to evaluate how motivation and engagement can be achieved. Pertaining to rehabilitation training, understanding the needs of the gamification application for enhancing patients' engagement in the therapy sessions is essential to set the foundations for this survey.

Previous research has attempted to explore the ways gamification can be effectively implemented in several rehabilitation settings (see studies by [1,28,29]). From the summaries elucidated in Appendix A, it was observed that gamification's main rehabilitation requirements are closely related to achieving gameful experiences, garnering intrinsic motivation, and yielding sustainable engagement for compliance with therapy sessions.

These generic needs aim to ensure that the gamification application is utilized as a part of the rehabilitation intervention that changes individual behavior in the rehabilitation process and, thus, adheres to the training for improvements in neurological and functional outcomes. For that, individualized and tailored game elements and techniques specifically designed for particular a rehabilitation domain would be the ideal solution.

## 5. Gamified Intervention in Rehabilitation

Gamified applications have been implemented in various ways. Generally, the implementation is designed to attain whichever gamification requirement is needed for a particular rehabilitation application. In this section, the types of gamification intervention used for rehabilitation are described. Subsequently, this section maps the relationship between the requirements and the types of intervention accordingly.

### 5.1. The Type of Gamified Intervention in Rehabilitation

In physical rehabilitation, various types of gamified-based interventions were employed in the selected articles, and the stroke was the domain with the highest frequency that utilized gamified-based interventions for rehabilitation training. Based on the initial survey, as summarized in Appendix A, the gamified-based interventions were categorized into robot-based system/Social robotics, virtual agent, game-based therapy, video game therapy, and Web-based intervention. Meanwhile, for cognitive rehabilitation, the most employed gamified-based interventions are serious game-based applications and web-based applications. The selection type of gamified-based interventions is based on their direct visibility to the end-user, i.e., the patient, and how easily one can operate them in the setting. Each of the intervention categories is described as follows:

1.  The robotics-based system/Social robotics is an external tool that is embedded physically in the rehabilitation to guide patients by giving instructions on how to play the gamified applications. Most importantly, the robot provides feedback on individual performances and achievements during and after gameplay (see studies by [11,15,30]). As social robotics, patients can invite the robot to play together and they can challenge each other in the gameplay. Additionally, social robotics is embedded as a game companion where the robot takes the role of a coach and motivator when performing rehabilitative exercises [15,31]. Other than the element of feedback, the gamified robot-based system uses levels. Each level has different rehabilitation tasks, and they are designed with increasing difficulty. The selected articles demonstrated that Social robotics could help improve individual motivation and engagement in rehabilitative therapies [11,30] and improvement in neurological deficits, particularly with motor functions [24].

2.  A virtual agent embedded during rehabilitation training shares a similar function to robotics. They serve as guides or playmates (companions) but they are virtually adopted in the application and not as a physical instrument attached to the application. As a guide, the implementation of a virtual agent acts as the virtual therapist that can track individual rehabilitation progress while assuming the role of a coach to prescribe appropriate exercises [5,29]. This approach was used to reduce face-to-face rehabilitation settings, thus decreasing consultation costs, and being less time-consuming and more logistically appealing [29]. A virtual agent is also implemented as an individual avatar—a self-representation in the gameplay [18,29,32–35]. The game consequences and feedback are visualized through the individuals' avatar in real-time [32,34]. Good feedback serves as a motivational drive, but poor feedback would either promote the progression element in training or demotivate patients, leading to dropouts from therapy sessions. Other than the element of feedback, the elements of level, dashboard, and avatar customization are also part of the gamified application. In the study by Rapp and Cena [18], replicating an individual virtually through an avatar is an excellent strategy to improve one's interest in managing personalized emotional care and psychological aspects during rehabilitation.

3.  Game-based therapy uses commercial off-the-shelf games adopted in gamified rehabilitation care. The game can be related to the individual condition or serve as a general game that is included in the rehabilitation environment. A game that imitates a real situation may demotivate patients at some point after playing the game [8,9]. A general game is included as a means to balance the environment, whereby it can evoke an immersive environment to the therapy sessions during rehabilitation [8]. Therefore, game-based therapy is instilled as a series of games connected to a mobile app [18,19,21,36,37], or a computer software application [32,38].

4.  Video game-based therapy uses commercial off-the-shelf games and implements mixed reality (virtual and augmented). This type of play is usually related to the exergame, one of the available kinds of games. For instance, the game has added devices or equipment, a Leap motion, a Wii balance board, and a tool for Microsoft Kinect. This approach is used as a support for exercising activity, fitness-related activities, or physiotherapy activities. Following Muñoz et al. [35], the Kinect motion sensor is used to detect an individual body movement during fitness training and these movements are reflected through their in-game avatar. This approach is similar to Manolova [3], but they used Leap motion as the motion sensor to detect hand movement during hand rehabilitation. In this approach, the video game is part of the gamified rehabilitation application. The patients play the related game, including the game elements such as the scoring and leaderboard element, for evoking gamefulness experiences during rehabilitation training.

5.  Web-based intervention is a gamified system that utilizes a web-based architecture for telerehabilitation. The architecture is either cloud-based [36] or server-based [39]. In the architecture, a series of games are implemented and personalized to individual needs. Personal analytics is utilized in a web-based system for demonstrating a summary of performances and achievements throughout the rehabilitation process.

6.  Social application is an application that allows an individual to socialize with other people while undergoing rehabilitation activities. Its use depends on whether the application is a games-based [26] or a web-based application [21]. By socializing virtually, individuals could collaborate or complete their rehabilitation performance, as well as discuss and exchange opinions about their condition. This type of rehabilitation application usually utilizes elements of social games, with game elements such as the avatar/agent, quest, leaderboard, and ranking.

*5.2. Mapping Intervention and Gamification Requirement*

The types of intervention show a diversity of gamification applications in accommodating rehabilitation needs. By scrutinizing the gamification requirements and categorizing the types of rehabilitation intervention, the relationship between them was mapped following the rehabilitation domain. The mapping is presented in Table 1. This table shows that the rehabilitative therapies for hand, upper limb, knee, body posture, gesture, and gait are the domains that have most-frequently implemented the gamified application in rehabilitation training. However, specific gamified applications are inclined towards the condition-based rather than the anatomical structure, in which neurological rehabilitation predominates. Psychological rehabilitation is the least-observed domain. Following the domain and the gamified intervention, gamification needs are distributed relatively equally, i.e., all are aimed at providing a gameful experience, providing an element that intrinsically evokes motivation, and having a sustainable engagement tool. All requirements are more likely to create adherence and compliance with rehabilitative therapy.

**Table 1.** Mapping gamified type of interventions with the requirement of gamification in rehabilitation training.

| Domain | Sub Domain | Gamified Type of Intervention | The Requirement/Needs | | | References |
|---|---|---|---|---|---|---|
| | | | Gameful Experiences | Sustainable Engagement | Intrinsically Motivation | |
| Rehabilitative therapies (physiotherapy) | Fostering Self-efficacy and encourage motor training/activities | Gamified application with a series of games | ⊘ | ⊘ | | [15,38,40] |
| | | Exergame-based training system | ⊘ | | ⊘ | [4,33,35,41] |
| | | Mobile Apps and Gamified application | | ⊘ | | [32,36,37] |
| | | Gamified robot-based system | | ⊘ | | [5,42] |
| | | Social Application and VR | | ⊘ | ⊘ | [26] |
| | Improve behaviour and health outcomes. | A Web-based intervention which includes gamification on physical activities | | | ⊘ | [25] |
| | Nurture self-learning motor recovery | Physical games related to therapy | | | ⊘ | [1] |
| Hand rehabilitation (upper-limb rehabilitation) | Motor recovery | Gamified application with mixed reality (VR/AR) assisted immersive in gamified rehabilitation | ⊘ | ⊘ | ⊘ | [22,43–45] |
| | | Gamified application with a series of adaptive games | | | ⊘ | [46] |
| | Improve muscle strength | Gamified application with series of games and motion sensors | ⊘ | ⊘ | | [3] |
| | Improve motor control | Gamified application with a series of games | | ⊘ | | [47] |
| | | Gamified robot-based system | ⊘ | | | [31] |
| General Stroke rehabilitation | Improve muscle strength | gamified robot-based system | | ⊘ | | [11] |
| | | Web based gamified system with games | | | ⊘ | [39] |
| | Improve motor coordination | Gamified application with a series of games | ⊘ | | ⊘ | [48] |
| | Improve movement and mobility | Games and smartphone sensors | | ⊘ | ⊘ | [20] |
| | | Gamified application with video games | ⊘ | ⊘ | | [49] |
| Neurological rehabilitation (Parkinson) | Improve movement and mobility | A video game—Virtual reality exergames Augmented Reality games | ⊘ | ⊘ | | [23,50] |
| Neurological rehabilitation | Reduce concussion | Mobile social gamified apps | | ⊘ | | [21] |
| | Improve quality of life | Gamified application with multiple roles of Software agent | | ⊘ | ⊘ | [29] |
| | | Video game using gamification approach | | ⊘ | ⊘ | [34] |
| | Nurture self-learning and adherence | Gamified application with a series of games | ⊘ | | ⊘ | [2,51] |
| Gait rehabilitation | Improve movement | Gamification approach using VR in training | | ⊘ | ⊘ | [52] |
| Psychology Rehabilitation | Improve quality of life | A gamified mobile e-Health application | | | ⊘ | [53] |

Meanwhile, the most-utilized rehabilitation intervention is gamified applications embedded with external devices such as robotics, mixed reality, and a series of games using Wii, Kinect, and motion sensors. For rehabilitation applications, external tools are considered as necessary provisions. These tools are used to (1) guide and facilitate the motor-rehabilitative therapies, (2) enhance individuals' motivation and engagement, and (3) provide a fun and enjoyable environment during debilitating health conditions such as stroke. The external devices are proven to be effective [16,22,33]. However, they require high expenditure, including on physical and software maintenance. For self-regulated rehabilitation training, these are less suitable as they lack accessibility for all user levels [46,52], unless the healthcare facilities or agencies support the implementation.

## 6. The Classification of Gamification Application in Rehabilitation

The identification and classification of the gamification application and its requirements in rehabilitation are the foundation of designing the required techniques for a gamified application. Subsequently, this section analyzes and classifies the game elements that are generally applied in a gamified application with the different selection sets of building blocks for gamification. These analyses are used to evaluate the users' motivation for, and engagement towards, rehabilitation training.

### 6.1. Gamification Building Blocks

Following the general list of game elements in gamification elucidated in Section 3.2, the building block selection that is mostly discussed and applied in rehabilitation training can be further summarized. These blocks consist of game elements that are usually designed together. This summarization works to provide ideas on the grouping of these elements that would affect the treatment and outcomes after rehabilitation training. The game elements building blocks are presented in Table 2. In the summary table, each selection block is mapped with a more or less similar gamification technique. The techniques are targeted to the individual experiences in gameplay. The techniques are the collection of targeted experiences based on the findings extracted from the selected articles for this study. These include motivational narratives, goal-oriented tasks, responsive feedback, rewards, fantasy, personalization, personal informatics, visual feedback, iterative feedback and evaluation, rewards, and personalized experience.

In the selected articles, it was found that game elements' distribution is mainly related to scoring (points), levels, and challenges, and the avatar, narrative, and feedback. Different building blocks were adopted to accommodate specific needs during rehabilitation training. For example, the feedback element is important in providing a quick response to the patients on how well they have worked on their exercises; hence, the ability for a self-regulated physio, in that this feedback is usually conveyed by an avatar [11,34]. The avatar is the agent that acts as the therapist, or caregiver, or self-representation. Other than the use of an avatar as the medium, feedback is also delivered through responsive interfaces. This visual feedback form helps patients progress further while enjoying the intervention during the therapy [1,11,20,34,39].

**Table 2.** Selection of building blocks used in rehabilitation gamification.

| Selection Block | References | Motivational Narratives | Goal-Oriented Tasks | Responsive Feedback | Rewards | Fantasy | Personalization | Personal Informatics | Visual Feedback | Iterative Feedback |
|---|---|---|---|---|---|---|---|---|---|---|
| Levels and challenges | [3,24,33,36,43] | ✓ | ✓ | | | | | | | |
| Scoring (points-based system), quests, and avatar | [32,35,40] | | | ✓ | ✓ | ✓ | | | | |
| Dashboard and levels | [4] | | | | | | ✓ | | | |
| Avatar (identification and personalization), narrative context, and feedbacks | [15,30] | ✓ | | | | | ✓ | | | |
| Avatar (identification and personalization), dashboard (analytics), levels, feedback, challenges, rewards, and narrative context | [11,20,34,50,54] | ✓ | | | | | | ✓ | ✓ | |
| Dashboard and feedback | [16,47] | | | | | | | ✓ | | ✓ |
| Leaderboards, score, levels, and challenges | [31,44,45,55] | | ✓ | | | | | ✓ | | |
| Points, badges, and medals (awards) | [21,25,49] | | | | ✓ | | | | | |
| Leaderboard, points, trophy, and rewards | [2] | | | | ✓ | | | | | |
| Badges, leaderboards, and levels | [51] | | | | ✓ | | | | | |
| Leaderboard, score (point), progression (tracking), star | [29] | | ✓ | | | | | | | |
| Feedback and rewards | [1,56] | | | | | | | | | ✓ |
| Levels, feedback, status, and challenge | [19,22] | | | | | | ✓ | | | |
| Feedback, progress, score, personalization, and dashboard (analytic) | [23,39] | | | | | | ✓ | | | ✓ |
| Points, badges, challenges, visualization of progress, narrative, goal settings, and visual progress | [53,57] | ✓ | | | | | | ✓ | ✓ | |

### 6.2. Classification of Gamification

In this paper, the gamification techniques were identified based on the most-used game mechanics in gamified applications utilized in rehabilitation training to motivate and sustain individual engagement in gameplay. It is argued that the game elements and mechanics are not the same. The term might be used interchangeably, and occasionally they are applied as gamification techniques. The gamification technique is most likely a game mechanic that differs from the game element. One strong justification is the fact that a game element is implemented as a selection block, as shown in Table 2 or, as explained in Section 3.2, when implemented individually. For example, a personalized experience is one of the gamification techniques that bring the individual immersion in the gameplay, which brings lived experiences to the person. As the design challenges require a gamified application that incorporates gamefulness, gameful interaction, and gameful design, a gamification technique that caters to these challenges should be addressed. Therefore, it is proposed to classify the game techniques based on the requirements or needs of gamification in rehabilitation, as mentioned in the earlier section: gameful experiences, intrinsic motivation, and sustainable engagement. This derived classification helps developers or designers, as well as researchers, deploy a suitable gamification application for rehabilita-

tion training. It emphasizes the importance of designing a gamification application based on rehabilitation needs.

Following the selected articles summarized in Appendix A, a stepwise approach was implemented to propose the classification of gamification techniques used in rehabilitation gamified applications. Basically, the process involved identifying the root of this classification, followed by the branches and nodes that evolved from it. Hence, the approach is as follows:

- First, the domain of rehabilitation was extracted. It was found that gamification in rehabilitation was related to the post-stroke condition, and the previous works were bounded in the domain of physiotherapy, neurology, and psychology. Thus, these domains developed as the general rehabilitation domains that been addressed for the implementation of gamification.
- Second, the common intervention tools were scrutinized and grouped within the same domain. However, the tools could also be cross-applied in other domains. In this category, the intervention group was proposed based on its suitability and effectiveness from previous works.
- Third, the gamified application was correlated with the gamification requirements usually designated when solving particular rehabilitation issues.
- Fourth, following the game elements' selection block listed in Table 3, the collection of extracted targeted techniques from the previous works was mapped with the gamification requirements in step 3. The techniques include motivational narratives, goal-oriented tasks, responsive feedback, rewards, fantasy, personalization, personal informatics, visual feedback, iterative feedback and evaluation, rewards, and personalized experience.
- Finally, following the techniques, the most-applied game elements are proposed, but they are not limited only to the list. The game elements can be applied following their suitability and purposes. A game element is proposed based on the commonly applied elements when using particular gamification techniques.

**Table 3.** Results of classification article testing.

| | Rehab Domain | Gamified Application | Gamification Requirements | Gamification Technique | Applied Game Element | Status |
|---|---|---|---|---|---|---|
| [32] | Physiotherapy | mobile & web system | Motivation Engagement | visual feedback Rewards | Feedback Scoring (Points-based system) | ⊘ |
| [4] | Physiotherapy | Exergame | Motivation and engagement | Personalize experiences | Narrative, challenges, avatar | ⊘ |
| [16] | Physiotherapy | Exergame | Motivation and engagement | Personal informatics | Dashboard feedback | ⊘ |
| [29] | Neurological diseases (dementia) | Web Application system | Motivation and engagement | Goal oriented task | Avatar, leaderboard, score | ⊘ |
| [34] | Neurological (Dysphonic) | Gamified application | Motivation and engagement | Visual feedback | Avatar, feedback, narrative context | ⊘ |
| [21] | Neurological (concussion) | Mobile apps | Motivation and engagement | Rewards | Points, badges, award | ⊘ |
| [18] | Psychological Rehabilitation | Gamified Web applicaton | motivation | Personal informatics | Progression, challenges, points | ⊘ |
| [53] | Psychological Rehabilitation | Gamified web application | Motivation | Motivational narratives Personal Informatics | Points, badges, challenges, progression | ⊘ |

A simplified gamification technique classification based on the rehabilitation domain and its requirement is illustrated in Figure 3.

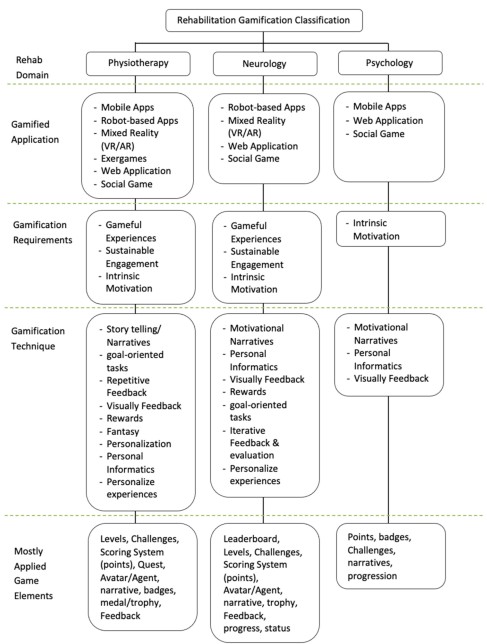

**Figure 3.** The Classification of gamification in rehabilitation.

Figure 3 shows the organization based on the implementation area. The gamified application has been used for intervention in rehabilitation, followed by gamifying requirements for rehabilitation, the gamification technique to accommodate the condition, and the game elements that are mostly used for the respective gamified application. Eight articles were selected randomly in order to comprehend the practicality of this classification. Hence, five variables related to the classification were tested. Table 3 shows the results. The result shows that all eight articles conformed to the classification. It is believed that a variety of approaches is warranted from a rehabilitation training perspective, and this classification is provided to guide the implementer, either the developers or the rehabilitation practitioners.

## 7. Challenges and Research Opportunity

Gamification for rehabilitation is not a new approach. As can be seen from the selected articles in Table 1, the gamification application has indeed been proven to be an effective intervention for enhancing individual participation in rehabilitation training. However, there are several possible research directions for future exploration.

First, few works focus on applications with the ability to support individuals other than the intended users, for example, the patient's caregiver. A caregiver is a person with great responsibility as an individual who personally cares for a disabled patient, either at home, at a hospital or in an institutional setting such as a nursing residence. Optionally, other individuals to consider are the physician and the therapist. Focusing on supporting these particular groups of people will promote a supportive and healthy environment, positive behavior, improved rehabilitation outcomes, and a cost-effective method of improving the quality of life. This recommendation has also been noted, discussed and developed by [2,11,58].

Second, an extensive evaluation of a larger group of patients is imperative for assessing the gamified application's feasibility, applicability, and practicality in a real environment. The evaluation should be conducted on a long-term clinical validation study (longitudinal studies) with focus group participants. The findings from such studies will validate the claims of the clinical effectiveness of using a gamified application in rehabilitation training. This recommendation has been expressed in research by [5,19,20,54]. By conducting rigor-

ous validation research, the deployment of a dynamic and robust gamified application in rehabilitation training is made possible.

Third, future works should focus on augmenting the gamification features. This type of application needs to be embedded with other gamification techniques and game elements; hence, improving the original application and expanding its features permits better implementation. A customizable game personalized to the individual condition is one of the innovative ways of retaining or enhancing individuals' motivation in going through rehabilitation training, as the road to recovery can be a long process. Furthermore, such improvement offers users the ability to adapt to greater flexibility in the gameplay environment. Social games in rehabilitation have been explored widely by [21,25,26]; however, integrating data analytics with social interaction features are also seen as exciting additions to the gamification application as this enables individual motivation to be boosted. Besides, similar work to that of [26] could be expanded with the various gamification techniques discussed in this paper. Fourth, a more comprehensive study to extend the application of gamification to other domains of rehabilitation would provide a considerable contribution to various perspectives of medical practices and the games industry.

With the above opportunities of rehabilitation gamification in research are the significant challenges of executing them successfully, as foreseen by technological perspectives and medical perspectives. One of the open challenges that have been continuously explored is the adoption of technology. Technology is highly dynamic and undergoes rapid changes, with more innovations invented periodically; gamification applications for health are no exemption. Choosing the right gamified application for health that is cost-effective while permitting low maintenance, and with the impetus for adaptability and flexibility for various rehabilitation domains is a significant challenge to the developer and the technology providers [22,41]. Hence, developing a robust application requires experts in this particular field of interest.

Moreover, with the introduction of a cyber-physical system in a smart-health application, one needs to anticipate a higher level of challenges in crafting a remotely available gameful experience. In addition to these technological changes, challenges can stem from the users themselves. These are the individuals who are directly utilizing the gamification application, either as patients, physicians, or therapists. They are going through a transitional phase from traditional, low-technology-based intervention to HCI-based, gamified applications. Adapting to the new mode of rehabilitation interventions demands readjustment needs. Strong teamwork and concerted effort from leading players in a rehabilitation team are imperative to counter such psychological needs.

## 8. Conclusions

Prior work on the gamification approach in improving individuals' motivation and engagement in rehabilitation training has proven to be an effective intervention. This paper aimed to analyze the gamified rehabilitation application and its requirements in rehabilitation, and to present the classification of gamification applications as a means to provide researchers with a basic guideline for crafting and deploying a gameful experience for recovery during rehabilitation. The analysis shows physical therapy as the domain with the highest frequency of utilizing the gamification approach for improving rehabilitation outcomes. For physical therapy, hand and upper limb rehabilitation are the areas that applied the gamification application most. Additionally, from various types of gamified applications, robot-based and mixed-reality adoptions are among the most popular ones. They are the usual interventions that satisfy the rehabilitation requirements, particularly in the domain of physiotherapy and neurological disease. Additionally, these analyses have led to the proposal and development of a classification of rehabilitation gamification to design and deploy the right application according to the rehabilitation needs. Commonly used techniques have been derived, such as the goal-oriented task, motivation narrative, iterative feedback, and personal informatics under the different types of applications and other kinds of tools. The previous gamified applications have shown a relatively similar

use of gamification techniques and selections of blocks of game elements in reference to the rehabilitation training. Notably, this research has surveyed the use of gamification applications in rehabilitation with specific insight into its capability for facilitating rehabilitation training. Despite the vast availability of such gamification applications, several research challenges remain to be explored for future studies. It can be concluded that this area of research has a lot to offer, especially in the health domain. Thus, as technology continues to develop and integrate into healthcare, further exploration has to be conducted for better rehabilitation outcomes.

**Author Contributions:** Conceptualization: N.M.T. and F.A.; data collection: N.M.T. and L.N.Y.; data analysis: N.M.T. and F.A.; writing—original draft preparation: N.M.T.; writing—review and editing: F.A. and A.G.; supervision: A.G. All authors have read and agreed to the published version of the manuscript.

**Funding:** This research received no external funding.

**Acknowledgments:** Authors thank University Malaysia Sabah for the research opportunity and financial support in the completion of this research.

**Conflicts of Interest:** The authors declare no conflict of interest.

## Appendix A

| No | Author, Year | Title | Problem Address | Rehab Domains | Gamification Application | Targeted Outcomes and Clinical Findings (If Relevant) |
|---|---|---|---|---|---|---|
| 1 | Buonocunto et al. [36] | A limb tracking platform for tel-erehabilitation | Addressing problems in motor-rehabilitative therapies for telerehabilitation care and providing solutions through a cyber-physical platform | Motor-rehabilitative | ReHapp, an android application—A Limb Tracking Platform for Tele-Rehabilitation | Outcome: Engagement to care |
| 2 | Chromy et al. [32] | DeskBalance: Novel Gamified System for Diagnosis and Treatment of Postural Stability | Introducing a gamified application for postural stability that provide individual diagnostic and therapeutic features. | Postural stability | DeskBalance, a gamified system—detect the patient's stabilization moves and presents them on the monitor as visual feedback | Outcome: Motivation and engagement to care |
| 3 | Segura et al. [15] | Playification: The PhySeEar case | Exploring the used of software agents in physiotherapy replacing the physiotherapist in playfulness environment | physiotherapy | "PhySeEar"—Agent-based in the gamification of rehabilitation therapy which acts as physiotherapist and playmate. | Outcome: Motivation and engagement to care |

| No | Author, Year | Title | Problem Address | Rehab Domains | Gamification Application | Targeted Outcomes and Clinical Findings (If Relevant) |
|---|---|---|---|---|---|---|
| 4 | González-González et al. [4] | Serious games for rehabilitation: Gestural interaction in personalized gamified exercises through a recommender system | How a recommender system can help the therapist to personalize gamified activities for the user | Physiotherapy (body gesture exercising) | TANGO: H as a personalized exercise designer and simulator for patients. | Outcome: Motivation and engagement to care |
| 5 | Korn & Tietz [33] | Strategies for playful design when gamifying rehabilitation. A study on user experience | Exploring how the game mechanics and which game mechanics have a successful implementation in gamification. Additionally, how those mechanics lead to improved motivation | Physiotherapy (motion/fitness train-ing/balance) | A gamified application implementing off-the-shelf games technology (Wii) | Outcome: Motivation and engagement to care |
| 6 | van der Kooij et al. [40] | Gamification as a sustainable source of enjoyment during balance and gait exercises | how the design and use of game elements help to develop individual's motivation over time and how enjoyment influences rehabilitation care, in particular, the motor performance | Motor-rehabilitative (physiother-apy) | A gamified application implementing a series of games (balance game and gait game) | Outcome: Motivation and engagement to care |
| 7 | Deacon et al. [16] | Can Wii balance? Evaluating a stepping game for older adults | Addressing balance control issues to older adults. Using off the shelf technology in the gamified system and providing iterative feedback to therapists and patients have helped increase participation and engagement. | Physiotherapy (balance control) | A gamified application implementing off-the-shelf games technology (Wii balance boards) | Outcome: Motivation and engagement to care |

| No | Author, Year | Title | Problem Address | Rehab Domains | Gamification Application | Targeted Outcomes and Clinical Findings (If Relevant) |
|---|---|---|---|---|---|---|
| 8 | Muñoz et al. [35] | Lessons Learned from Gamifying Functional Fitness Training Through Human-Centered Design Methods in Older Adults | They were designing a beneficial application for fitness training in older adults using game elements as motivation. Lack of synchronization between game preferences, technology, and involvement for users has resulted in a poor design application. | Physiotherapy (fitness training) | Gamified application using Kinect motion sensor to detect body movements and manipulate avatar in a game. | Outcome: Motivation and engagement to care |
| 9 | Stanmore et al. [41] | The effectiveness and cost-effectiveness of strength and balance Exergames to reduce fall risk for people aged 55 years and older in the UK assisted living facilities: A multi-centre, cluster randomized controlled trial | They explore the effectiveness of developed exergame gamified application in improving balance for older adults how the game can maintain body functions and reduce the risk of falling. | Physiotherapy (balance control) | Gamified application: OTAGO/FaME-based strength and balance Exergame for improving balance, maintaining the function, and reducing falls | Outcome: Motivation and engagement to care |
| 10 | Kappen et al. [28] | Gamification through the Application of Motivational Affordances for Physical Activity Technology | Motivational affordances design and application or game elements that promotes participation in physical activities in particular to a different group of users (age) | General Rehabilitation | No gamified application but used Exercise Motivations Inventory-2 scale (EMI-2) to survey gamification application in rehabilitation | Outcome: Motivation affordances in the gamification of therapy |

| No | Author, Year | Title | Problem Address | Rehab Domains | Gamification Application | Targeted Outcomes and Clinical Findings (If Relevant) |
|---|---|---|---|---|---|---|
| 11 | Manalova [3] | Application for Hand Rehabilitation Using Leap Motion Sensor Based on a Gamification Approach Application for Hand Rehabilitation Using Leap Motion Sensor Based on a Gamification Approach | Lack of research exploration and direction of hand gesture recognition using a gamification approach | Physiotherapy (Hand motion) | Gamified application: using a series of games and leap motion sensor | Outcome: Motivation and engagement to participate more in the care |
| 12 | Afyouni et al. [5] | Rehabot: Gamified virtual assistants towards adaptive telere-habilitation | Challenges in traditional therapy lead to attempts to use technologies in physical rehabilitation that can guide the patient in completing their therapy session. | Body Therapy (gesture) | Rehabot—a gamified robot-based system; virtual assistants to guide the patient through the different sets of gestures required to complete the session | Outcome: Motivation and engagement to care |
| 13 | Afyouni et al. [54] | A therapy-driven gamification framework for hand rehabilitation | Lack of research in facilitating a therapy-based hand gestures application to navigate movement. Also, less application and framework that address ab enriching user experiences in a gameful environment for individuals with hand disabilities. | Therapeutic hand exercises (gesture) | e-health adaptive serious games using leap motion controller to facilitate and encourage individual with hand disabilities | Outcome: significant improvement in patient engagement |
| 14 | Fotopoulos et al. [44] | Gamifying Motion Control Assessments Using Leap Motion Controller | Incorrect motor patterns through exercising have a considerable impact on individual motivation and experiences in therapy activities | Physiotherapy (Hand motion) | Gamified application using the virtual 3D game environment and leap motion controller for interaction | Outcome: Motivation and engagement to participate in an exercise session |

| No | Author, Year | Title | Problem Address | Rehab Domains | Gamification Application | Targeted Outcomes and Clinical Findings (If Relevant) |
|---|---|---|---|---|---|---|
| 15 | Colomer et al. [43] | Effect of a mixed reality-based intervention on arm, hand, and finger function on chronic stroke | A low-cost intervention for motor recovery after stroke is required to transform conventional tools into a practical intervention | Hand rehabilitation | A gamified system that used augmented reality in motor and ADL training | Outcome: useful and motivating for rehabilitation Findings: Significant improvement in arm function (Wolf Motor Function Test, $p < 0.01$) and finger dexterity measured (Box and Blocks Test, $p < 0.01$; Nine Hole Peg Test, $p < 0.01$) |
| 16 | Eizicovits et al. [31] | Robotic Gaming Prototype for Upper Limb Exercise: Effects of Age and Embodiment on User Preferences and Movement | The requirement to understand the effectiveness of human-robot interaction in rehabilitation | Upper limb rehabilitation | A robotic system; Robot-assisted therapy as a partner on everyday task and using a gamified approach | Outcome: human-robot interactions Motivate an individual's engagement |
| 17 | Allam et al. [25] | The Effect of Social Support Features and Gamification on a Web-Based Intervention for Rheumatoid Arthritis Patients: Randomized Controlled Trial | Lack of studies in the usefulness of online social support particularly the gamification on individuals behaviour and health conditions | Hand rehabilitation | a Web-based application with gamification on physical activities that provides patient-centred health education | Outcome: promising individual's behavioural and health outcomes. Findings: Increased Physical activity ($B = 3.39$, $p = 0.02$) |
| 18 | Melero et al. [45] | Upbeat: Augmented Reality-Guided Dancing for Prosthetic Rehabilitation of Upper Limb Amputees | Unsuccessful rehabilitation rate for upper limb amputees because of dissatisfaction and rejection in rehabilitation experiences | Upper limb rehabilitation | Gamified application with augmented reality (AR) dance game | Outcome: Motivation and engagement in care |

| No | Author, Year | Title | Problem Address | Rehab Domains | Gamification Application | Targeted Outcomes and Clinical Findings (If Relevant) |
|---|---|---|---|---|---|---|
| 19 | Stuart [2] | Exercise as therapy in congenital heart disease—A gamification approach | Lack of research in congenital heart rehabilitation. More evidence of beneficial treatment in congenital heart treatment is needed particularly in adherence to the rehabilitation activities | Congenital heart | Gamified method—uses off-the-shelf games (Nike+ fuelband) | Outcome: behavioural motivation (commitment to an exercise) Findings: Physical activity (measured with fuel points, a scoring system in the game) improved by 96% after three months. |
| 20 | Dithmer et al. [51] | "The Heart Game": Using Gamification as Part of a Telere-habilitation Program for Heart Patients | Low participation of heart patient in rehabilitation treatment and care due to many problems and this include the lack of information and motivation | Cardiac rehabilitation | The Heart Game—Web-based Gamification of the tele-rehabilitating process of heart disease patients | Outcome: Motivation and engagement in care |
| 21 | Polak et al. [11] | Novel gamified system for post-stroke upper-limb rehabilitation using a social robot: Focus groups of expert clinicians | Repetitive practice of rehabilitation is a great challenge to the therapist due to limited therapy time with the patient. Besides, the patients have to be highly motivated to engage with the repetitive tasks | Post-stroke upper limb | gamified robot-based system (using a socially assistive robot -SAR) | Outcome: Motivation and socially engaged in care |
| 22 | Vallejo et al. [29] | An agent-based approach to physical rehabilitation of patients affected by neurological diseases | A neurological disorder is a global challenge, and understanding their technology requirement is essential to improve the patient's quality of life. For that, agent technology may help to improve rehabilitation care. | Neurological diseases (stroke, dementia, ABI) | A gamified application implementing software agent that can play different characters to support the rehabilitation process | Outcome: Motivation and engagement in care |
| 23 | Janssen et al. [1] | Gamification in physical therapy: More than using games | Applying gamification design therapy session is argued to provide a positive physical and cognitive behavioural pattern to the recovery process | Therapeutic | Gamification approach that uses physical games related to therapy | Outcome: Behavioural patterns for treatment and recovery |

| No | Author, Year | Title | Problem Address | Rehab Domains | Gamification Application | Targeted Outcomes and Clinical Findings (If Relevant) |
|---|---|---|---|---|---|---|
| 24 | Lv et al. [34] | Clinical feedback and technology selection of game-based dysphonic rehabilitation tool | Dysphonic therapy is archived through the different pitch estimation of the patient's voice. | Neurological (Dysphonic rehabilitation) | Video game using gamification approach by providing visual feedback for voice exercises | Outcome: Motivation and engagement in care |
| 25 | Ferreira et al. [20] | Gamification of stroke rehabilitation exercises using a smartphone | The patient's compliance and motivation for stroke therapy are deficient. | Stroke rehabilitation | Gamification approach using games and phone sensors as a controller | Outcome: Motivation to exercise and compliance in therapy |
| 26 | Elor et al. [22] | Project Star Catcher: A novel immersive virtual reality experience for upper limb rehabilitation | Evidence of the immersive virtual reality that has a high potential to motivate the patients to go through the recovery process | Upper limb rehabilitation | Project Star Catcher (PSC)—an immersive virtual reality (VR) game in gamified rehabilitation | Outcome: Motivation and engagement in care |
| 27 | Jung et al. [39] | Ubiquitous gamification framework for stroke rehabilitation treatment based on the web service | Patient's motivation easily breakdown and delay participation in the rehabilitation program | Stroke rehabilitation | A web gamification framework with individualized game | Outcome: Motivation and engagement in care |
| 28 | Worthen-Chaudhari et al. [21] | Reducing concussion symptoms among teenage youth: Evaluation of a mobile health app | Youth are at risk of concussion. Mobile health platform is a creative way to connect youth with the concussion problem, particularly in tracking symptom, give treatment and provide support. | Neurological (concussion) | a mobile health application that includes gamification and elements of social game | Outcome: Motivation and engagement in care Findings: symptoms improved more for the experimental than for the active control cohort (U = 18.5, $p$ = 0.028, r = 0.50) |

| No | Author, Year | Title | Problem Address | Rehab Domains | Gamification Application | Targeted Outcomes and Clinical Findings (If Relevant) |
|---|---|---|---|---|---|---|
| 29 | Taherian et al. [56] | Are we there yet? Evaluating commercial-grade brain-computer interface for control of computer applications by individuals with cerebral palsy | The used of brain-computer interaction as assistive technology for the recovery process of individuals with cerebral palsy | Neurological (Cerebral Palsy) | A gamified application that uses EEG-based visual feedback | Outcome: Promoting motivational encouragement in treatment Findings: variable inconclusive results in the ability to produce two distinct EEG patterns. |
| 30 | Gauthier et al. [49] | Video Game Rehabilitation for Outpatient Stroke (VIGoROUS): protocol for a multicenter comparative effectiveness trial of in-home gamified constraint-induced movement therapy for rehabilitation of chronic upper extremity hemiparesis | Constraint-Induced Movement therapy clinical comparative effectiveness trial of in-home gamified therapy has not been conducted yet. | Stroke rehabilitation (hemiparesis) | Game-based exercise using commercially available games in the market and gamified home-based monitoring systems | Outcome: promising individual's behavioural and health outcomes. |
| 31 | O'Neil et al. [50] | Virtual Reality for Neuroreha-bilitation: Insights From 3 European Clinics | The clinical evidence of the effectiveness of virtual reality for motor impairment treatment is sparse | Neurological (Parkinson) | Virtual reality assisted rehabilitation for Parkinson's neurorehabilita-tion and gamified home-based tel-erehabilitation application | Outcome: promising individuals behavioural and health outcomes. |

| No | Author, Year | Title | Problem Address | Rehab Domains | Gamification Application | Targeted Outcomes and Clinical Findings (If Relevant) |
|---|---|---|---|---|---|---|
| 32 | Yunusova et al. [59] | Game-Based Augmented Visual Feedback for Enlarging Speech Movements in Parkinson's Disease | The feasibility of using augmented visual feedback to speech rehabilitation treatment on hypokinesia due to Parkinson. | Neurological (Parkinson-speech therapy) | Gamified application for speech therapy via Augmented Visual Feedback (AVF) | Outcome: promising individuals behavioural and improving health outcomes. Findings: 8/9 participants increased their articulatory working space to a greater degree. |
| 33 | Chen et al. [19] | Breeze: Smartphone-based Acoustic Real-time Detection of Breathing Phases for a Gamified Biofeedback Breathing Training | Slow-paced feedback that guides breathing has shown an improvement in cardiac functioning. | Cardiac rehabilitation | A mobile app application uses Gamified biofeedback-guided for breathing training | Outcome: promising individuals behavioural and improving health outcomes. Findings: User's significantly lower breathing rates during the Breeze training (b = −6.27, $p < 0.001$) and the active control condition (b = −3.24, $p < 0.001$). |
| 34 | van Dooren et al. [53] | Reflections on the design, implementation, and adoption of a gamified eHealth application in youth mental healthcare | How game design and techniques are applied for a gamified e-health application | Psychological rehabilitation | A gamified e-Health application for youth mental healthcare | Outcome: Change in behaviour and motivation |
| 35 | Rapp & Cena [18] | Personal informatics for everyday life: How users without prior self-tracking experience engage with personal data | Self-tracking or self-monitoring becomes an issue when dealing with a lot of health personal record | Psychological rehabilitation | A mobile app that includes gamification and personal informatics | Outcome: Change in behaviour and motivation |

| No | Author, Year | Title | Problem Address | Rehab Domains | Gamification Application | Targeted Outcomes and Clinical Findings (If Relevant) |
|---|---|---|---|---|---|---|
| 36 | Pinto et al. [46] | Adaptive Gameplay and Difficulty Adjustment in a Gamified Upper-Limb Rehabilitation | Patient lack of motivation in physical rehabilitation treatment and care | Upper-limb rehabilitation | a gamified application uses Kinect game for the home-based rehabilitation system | Outcome: promote motivation and engagement |
| 37 | Ozgur et al. [42] | Gamified Motor Training with Tangible Robots in Older Adults: A Feasibility Study and Comparison with the Young | Sustaining motor skills and functions is important. Assessing the effectiveness of robotic for upper limb motor in gamify environment is minimal | Upper-limb rehabilitation | Gamified Motor Training with the Cellulo, a Tangible Robots system | Outcome: promising individuals behavioural and improving health aging Finding: Elderly participants were able to improve their game performance over time ($t(874) = 2.97$, $p < 0.01$) |
| 38 | Kern et al. [52] | Immersive virtual reality and gamification within procedurally generated environments to increase motivation during gait rehabilitation | Gait rehabilitation required a tool that increases motivation which is achieved by using virtual reality technology | Gait Rehabilitation | Gamification approach in the immersive VR gait training | Outcome: Motivation and engagement in care |
| 39 | Dobosz et al. [37] | Gamification of cognitive rehabilitation | Improving the quality of brain functions through different cognitive processes | Cognitive Rehabilitation | "RehaMob"— A mobile application for e-health that used gamification for the operation of rehabilitation | Outcome: A positive influence in care |
| 40 | Chung & Ching [38] | Developing and Evaluating Creativity Gamification Rehabilitation System: The Application of PCA-ANFIS Based Emotions Model | Factors that influence rehabilitation achievement after knee replacement | Physiotherapy (knee) | An emotion model-based game rehabilitation system, which combines virtual reality (VR) and motion capture technology | Outcome: Motivation and engagement in care |

| No | Author, Year | Title | Problem Address | Rehab Domains | Gamification Application | Targeted Outcomes and Clinical Findings (If Relevant) |
|---|---|---|---|---|---|---|
| 41 | Carneiro et al. [47] | A gamified approach for hand rehabilitation device | Searching for an improved recovering approach for hand rehabilitation | Hand rehabilitation | A gamified approach attached with a set of therapeutic games and implementing augmented feedback | Outcome: Motivation, engagement and therapy effectiveness |
| 42 | Arlati, S. et al. [26] | A Social Virtual Reality-Based Application for the Physical and Cognitive Training of the Elderly at Home | Improve treatment adherence and motivation to do home self exercises | Physical and Cognitive Training | A virtual-reality social games with SocialBike application for elderly users home rehabilitation | Outcome: Motivation, engagement |

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
