# Peer review of "A Survey on Gamification for Health Rehabilitation Care: Applications, Opportunities, and Open Challenges"

_information, doi:10.3390/info12020091_

Round 1

Reviewer 1 Report

The paper investigates gamification techniques for rehabilitation applications. The topic presented in this paper is interesting and fits the scope of this journal.

The Introduction provides the scientific context for this survey, but it lacks in providing a definition of "gamification techniques" - a necessary definition in my opinion, since this concept is relevant also for the rest of the paper.

Some concerns regarding Sect.2: What are the methods adopted to conduct the survey? How did the authors select the papers? Some methods are described in line 188, but it is unclear why the authors selected that specific year range, and it is not expressed which keywords were adopted to conduct the database research (and where they were searched: title, abstract, article body). What kind of publications were considered in this survey? Journal papers, conference papers, reports, etc.? I would suggest to move and improve the methods in a dedicated subsection of Sect. 2. Although the paper's results are based on a survey, I would argue that 41 research articles may not be enough to draw general conclusions (especially if the methodology adopted to extract this 41 papers out of an unspecified number of papers originally retrieved is not detailed).

Sect. 2.2 opens with a definition of the practical fields in which rehabilitation has proved useful: among these, the authors acknowledge "daily living activities". If these activities are Katz's ADLs, they should be mentioned properly and a reference to Katz' work (1983) should be mandatory.
This Sect. requires a deep revision of English, as sometimes concepts are not clearly expressed nor properly linked among them (lines 163 - 170): this may jeopardize the proper understanding of authors' presentation.
Lines 186 - 187 start to provide a hint of a classification: however, the classes here illustrated are not consistent. Mobile-based, AR & VR, Web-based are classes focused on the technological means; exergames and serious games may use one or more of the technologies mentioned in the previous sentence (in other words, these two classes are more focused on the function of the exercise and should not be appear in the same list of the others).
I would suggest also to take into account another dimension in the classification: some rehabilitation games can be also played by more than one individual at the same time, thus adding a social aspect to rehabilitation (see for example: 10.3390/s19020261).

Sect. 3 delves into the requirements for implementing gamification in rehabilitation.
Lines 242 243 refer to a "conventional method": what is it?
In 3.1 lines 250 - 252 introduces the three elements considered as requirements: however, they are defined only in lines 263 - 271 - I suggest to first define the three terms, then discuss them.
In Sect. 3.2 the various types of gamified interventions are addressed. The whole subsect. 3.2 seems to deal more with means-related classification, other than requirements necessary for gamified applications for rehabilitation.
Sect. 3.3 is interesting; however it would be even more interesting to split each Domain (first column of Table 1) into a set of specifics needs (e.g.: for Stroke rehabilitation, we may have: improve muscle strength, improve coordination, improve mobility [for individuals on a wheelchair - see i.e. 10.3233/WOR-182829], etc.) and for each of these specific needs it would be interesting to find out if gamification approaches and technologies can be somehow related.

Sect. 4.1 opens with a list of game elements, properly defined. However, those elements have already been mentioned throughout the previous sections; again, I'd suggest to present definitions first, then discuss the various elements. Moreover, since Sect. 4 should be dedicated to the presentation of the classification, I was expecting to have acquired all the elements to properly understand it before this Sect.; i.e., I was not expecting another list of definitions.
What methodology was used to produce the categories populating authors' classification? In other words, how did the authors performed the task described in lines 518 - 519? Since the result of the survey is presenting a classification - which is expected to be reused and shared by experts or other relevant stakeholders - I would suggest to delve more into the methods used to generate the classification. Also, once the classification is presented, it could be interesting to "see it in action" - trying to use it to classify a couple of applications (taken from the literature already presented) to illustrate the functioning of your model.

Section 5 illustrates future lines of research; however it is limited to highlighting well-known lines of research. In line 556 - 567 mention the social element, but this direction has been widely investigated (see: 10.3390/s19020261)- Lines 568 - 576 underline a challenge that is quite obvious and addressed in many research papers.

--- General remarks on the paper: ---

The entire manuscript would benefit from a grammar revision, especially wrt compound tenses and their use. The whole text is scattered with unclear sentences that require rephrasing (e.g.: line 38, line 44, line 59, lines 71-73, line 106, etc.). I would like to point out that this issue can hinder the comprehension of many paragraphs, therefore I suggest authors to proofread the manuscript.

As emerged from the comments above, the manuscript could benefit from a reorganization of the contents: at first, the methodologies should be clearly presented; second, all the relevant elements that are going to be used to organized the classification should be presented; third, the analysis conducted and finally the presentation of the classification - followed by Sect. 5 - "Challenges and research opportunities".

Also, a recent literature review (10.1016/j.jbi.2017.05.011) covers the scope of this manuscript: it could be advised to highlight what is different (the classification). Once again, the topic is interesting, but widely investigated by different perspective - therefore this paper must be somehow unique.

I suggest the authors to revise the manuscript.

--- Manuscript presentation notes: ---
Authors' affiliations could be unified (please see the Journal's guidelines for Authors).
Also, the PDF version I was able to download shows some minor issues with document margins.

Author Response

Dear Reviewer,

Thank you for the comments. We appreciate it and we have made amendment based on the comments. We attached the reply to comments document. 

Thank you.

Reviewer 2 Report

The authors have carried out a review of papers describing the application of game-design elements and techniques at medical rehabilitation. After introducing the concept of gamification and explaining all the gamification techniques that can be used for rehabilitation care, the authors present a classification of the papers. Finally, challenges and opportunities for future research have been highlighted.

In my opinion, this review paper seems to be comprehensive and deals with a highly topical subject. It can be of interest for experts and researchers. However, there are some issues that must be addressed:

-  The paper needs proofreading. Most of the text does not read well and there are some grammatically incorrect sentences.

- References must be rearranged, so that they are listed in the order in which they appear in text.

Author Response

Dear Reviewer,

Thank you for your comments. My responses are as follow;

Comment 1;

The paper needs proofreading. Most of the text does not read well and there are some grammatically incorrect sentences.

Reply;

We submitted to proofreader and the latest one is proofread.

Comment 2;

References must be rearranged, so that they are listed in the order in which they appear in text.

Reply;

We rearranged the references as suggested.

Thank you.

Round 2

Reviewer 1 Report

The Authors have deeply reviewed the paper and made great efforts to meet my comments and suggestions. The Introduction is now more complete and accessible.

Section 2 now encompasses the methodology used for the survey and an in-depth description of the literature retrieval process; it is a very complete and detailed work. 

The definitions and details provided in Sections 3 and 4 allows readers to always have a conceptual grasp of everything being described or commented in these two essential sections. 

Adding the social application (Section 5) completes the overview.

Section 6 is now enriched by detailing the steps followed to get the classification, and an interesting first attempt to "test" the classification was also conducted (lines 595-601).  

The presentation of the manuscript has improved, together with the quality of English language. 

I think this work is ready for publication on MDPI- Information.

Reviewer 2 Report

The authors have addressed all my comments and suggestions. I think the paper has been improved and can be published.